

**The influence of internal variability on Earth's energy balance framework and implications for**
**estimating climate sensitivity**
Andrew E. Dessler[1]*, Thorsten Mauritsen[2], Bjorn Stevens[2]
[1] Dept. of Atmospheric Sciences, Texas A&M University, College Station, TX 77843
[2] Max Planck Institute for Meteorology, Bundesstraße 53, 20146 Hamburg, Germany
*Correspondence to: adessler@tamu.edu, 979-862-1427
Keywords: Climate sensitivity, climate variability, energy balance





**Abstract:** Our climate is constrained by the balance between solar energy absorbed by the
Earth and terrestrial energy radiated to space.  This energy balance has been widely used to
infer equilibrium climate sensitivity (ECS) from observations of 20th-century warming.  Such
estimates yield lower values than other methods and these have been influential in pushing
down the consensus ECS range in recent assessments.  Here we test the method using a 100-
member ensemble of the MPI-ESM1.1 climate model simulations of the period 1850-2005 with
known forcing.  We calculate ECS in each ensemble member using energy balance, yielding
values ranging from 2.1 to 3.9 K.  The spread in the ensemble is related to the central
hypothesis in the energy budget framework: that global average surface temperature
anomalies are indicative of anomalies in outgoing energy (either of terrestrial origin or reflected
solar energy).  We find that assumption is not well supported over the historical temperature
record in the model ensemble or more recent satellite observations.  We find that framing
energy balance in terms of 500-hPa tropical temperature better describes the planet's energy
balance.



**The problem**
When an energy imbalance is imposed, such as by adding a greenhouse gas to the atmosphere,
the climate will shift in such a way to eliminate the energy imbalance.   This process is
embodied in the traditional linearized energy balance equation:

27       $R = F + \lambda\ T_s$                                                  (1)

where the forcing F is an imposed energy imbalance, $T_S$ is the global average surface
temperature, and $\lambda$ relates changes in $T_S$ to a change in net top-of-atmosphere (TOA) flux
(Dessler and Zelinka, 2014).  R is the resulting TOA flux imbalance from the combined forcing
and response.  All quantities are deviations from an equilibrium base state, usually the pre-
industrial climate. Equilibrium climate sensitivity (hereafter ECS, the equilibrium warming in
response to a doubling of $CO_2$) is equal to $-F_{2xCO2}/\lambda$, where $F_{2xCO2}$ is the forcing from doubled
$CO_2$.
Many investigators (e.g., Gregory et al., 2002; Annan and Hargreaves, 2006; Otto et al., 2013;
Lewis and Curry, 2015; Aldrin et al., 2012; Skeie et al., 2014; Forster, 2016) have used Eq. 1
combined with estimates of R, F, and $T_s$ to estimate $\lambda$:

38       $\lambda = \Delta(R-F)/\Delta T_s$                                      (2)

where $\Delta$ indicates the change between the start of the historical period (usually the mid to late
nineteenth century) and a recent period.  These calculations result in values of $\lambda$ near
-2 W/m$^2$/K and appear to rule out ECS larger than ~4 K (Stevens et al., 2016).  The substantial
likelihood of an ECS below 2 K implied by these calculations led the IPCC Fifth Assessment
Report to extend their lower bound on *likely* values of ECS to 1.5 K (Collins et al., 2013).
We test this energy balance methodology through a perfect model experiment consisting of an
analysis of a 100-member ensemble of runs of the MPI Earth System Model, MPI-ESM1.1.  This
is the latest coupled climate model from the Max Planck Institute for Meteorology and consists
of the ECHAM6.3 atmosphere and land model coupled to the MPI-OM ocean model. The
atmospheric resolution is T63 spectral truncation, corresponding to about 200 km, with 47





vertical levels, whereas the ocean has a nominal resolution of about 1.5 degrees and 40 vertical
levels. MPI-ESM1.1 is a bug-fixed and improved version of the MPI-ESM used during CMIP5
(Giorgetta et al., 2013) and nearly identical to the MPI-ESM1.2 (Mauritsen et al., 2018) model
being used to provide output to CMIP6, except that the historical forcings are from the MPI-
ESM.
Each of the 100 members simulates the years 1850-2005 (Fig. 1) and use the same evolution of
historical natural and anthropogenic forcings.  The members differ only in their initial
conditions —each starts from a different state sampled from a 2000-year control simulation.
We calculate effective radiative forcing F for the ensemble by subtracting top-of-atmosphere
flux R in a run with climatological sea surface temperatures (SSTs) and a constant pre-industrial
atmosphere from average R from an ensemble of three runs using the same SSTs but the time-
varying atmospheric composition used in the historical runs (Hansen et al., 2005; Forster et al.,
2016).  The three-member ensemble begins with perturbed atmospheric states.  We estimate
$F_{2xCO2}$ using the same approach in a set of fixed SST runs in which $CO_2$ increases at 1% per year,
which yields a $F_{2xCO2}$ value of 3.9 W/m$^2$.
We calculate λ using Eq. 2 for each ensemble member, producing values ranging from -1.88 to
-1.01 W/m$^2$/K, with an ensemble average of -1.43 W/m$^2$/K (Fig. 2a).  In this calculation, Δ(R-F)
and $\Delta T_S$ are the average difference between the first and last decade of each run. We also
calculate ECS = $-F_{2xCO2}$/λ for each ensemble member, producing values ranging from 2.08 to
3.87 K (Fig. 2b), with an ensemble average of 2.76 K.
With respect to precision of the estimates, our analysis shows that λ and ECS estimated from
the historical record can vary widely simply due to internal variability. Given that we have only
a single realization of the 20[th] century, we should not consider estimates based on the historical
period to be precise — even with perfect observations. This supports previous work that also
emphasized the impact of internal variability on estimates of λ and ECS (Huber et al., 2014;
Andrews et al., 2015; Zhou et al., 2016; Gregory and Andrews, 2016).



Previous researchers have questioned whether the historical record provides an accurate
measure of $\lambda$ and ECS, and we can check this by comparing the ensemble average values to ECS
estimates from a 2xCO$_2$ run of the MPI-ESM1.2, which is physically very close to MPI-ESM1.1
and the changes between the MPI-ESM1.1 and MPI-ESM1.2 are not believed to be important
for its climate sensitivity.  An abrupt 2xCO$_2$ run yields an ECS of 2.93 K in response to an abrupt
doubling of CO$_2$ (estimated by regressing years 100-1000 of a 1000-year run) — 6% larger than
the ensemble average. This is in line with the 10% difference in ECS estimated by Mauritsen and
Pincus (2017) to arise from the average CMIP5 model time-dependent feedback, but is smaller
than suggested in other recent studies of ECS in transient climate runs (e.g., Armour, 2017;
Proistosescu and Huybers, 2017).
Thus, there are a number of issues that need to be considered when interpreting estimates of $\lambda$
and ECS derived from the historical period.  In addition to the precision and accuracy issues
discussed above, it also includes the large and evolving uncertainty in forcing over the 20[th]
century (Forster, 2016), different forcing efficacies of greenhouse gases and aerosols (Shindell,
2014; Kummer and Dessler, 2014), and geographically incomplete or inhomogeneous
observations (Richardson et al., 2016).
**Why are estimates using the traditional energy balance approach imprecise?**
In this section, we explain the physical process by which internal variability leads to the large
spread in $\lambda$ and ECS estimated from the ensemble.  We begin by observing that Eqs. 1 and 2
parameterize R-F in terms of T$_S$.  In model runs with strong forcing driving large warming, such
as abrupt 4xCO$_2$ simulations, there is indeed a strong correlation between these variables (e.g.,
Gregory et al., 2004).  However, because R-F in such runs is dominated by a monotonic trend,
correlations will exist with any geophysical field that also exhibits a monotonic trend, regardless
of whether there is a physical connection between the fields. Thus, one should not take the
correlation between R-F and T$_S$ in these runs as proving causality.
If T$_S$ is a good proxy for the response R-F, we would expect to also see a correlation in
measurements dominated by interannual variations. Observational data allow us to test this





hypothesis.  We use observations of R from the Clouds and the Earth's Radiant Energy System
(CERES) Energy Balanced and Filled product (ed. 4) (Loeb et al., 2009), which cover the period
March 2000 to July. 2017. Our sign convention throughout the paper is that downward fluxes
are positive.  Temperatures come from the European Centre for Medium Range Weather
Forecasts (ECMWF) Interim Re-Analysis (ERAi) (Dee et al., 2011).  We assume forcing changes
linearly over this time period and account for it by detrending $\Delta R$ and $\Delta T$ anomaly time series
using a linear least-squares fit to remove the long-term trend.
These data show that $\Delta R$ is poorly correlated with $\Delta T_s$ in response to interannual variability (Fig.
3a), as has been noted many times in the literature; see, e.g., Sect. 5 of Forster (2016).  In
particular, the low correlation coefficient tells us that $\Delta T_S$ explains little of the variance in $\Delta R$.
Using explicit estimates of forcing or other temperature datasets (e.g., MERRA-2) yield the
same result.
GCMs that submitted output to the 5[th] phase of the Coupled Model Intercomparison Project
(CMIP5) (Taylor et al., 2012) also show this poor correlation.  To demonstrate this, we have
calculated the correlation coefficient between $\Delta T_S$ and $\Delta R$ in CMIP5 pre-industrial control runs
(these are runs for which forcing F = 0).  To facilitate comparison with the CERES data, as well as
avoid any issues with long-term drift in the control runs, we break each run into 16-year
segments and calculate the correlation coefficient of monthly anomalies of $\Delta R$ and $\Delta T_S$ for each
segment. Fig. 4 shows that the correlation between $\Delta R$ and $\Delta T_S$ in the models is similar to that
from the CERES analysis.
Recent work provides an explanation: the response of $\Delta(R-F)$ to a particular $\Delta T_S$ is determined
not only by the global average magnitude, but also by the pattern of warming (Armour et al.,
2013; Andrews et al., 2015; Gregory and Andrews, 2016; Zhou et al., 2016, 2017; Andrews and
Webb, 2018). During El Nino cycles that dominate the observations in Fig. 3, the spatial pattern
of warm and cool regions changes, leading to responses in $\Delta(R-F)$ that do not scale cleanly with
$\Delta T_S$ — something Stevens et al. (2016) refer to as "pattern effects"




To demonstrate how this also generates the spread in $\lambda$ in the model ensemble (Fig. 2a), we
calculate the local response $\lambda_r$ in three equal-area regions (90°S-19.4°S, 19.4°S-19.4°N, 19.4°N-
90°N).  We define $\lambda_r$ as the regional analog to $\lambda$ (Eq. 2):

131        $\lambda_r = \Delta(R\text{-}F)_r / \Delta T_{S,r}$                                    (3)

where the "r" subscript indicates a regional average value.
We find that $\lambda_r$ varies between the regions (Fig. 5). This means that different ensemble
members with similar global average $\Delta T_S$ but different patterns of surface warming produce
different values of global average $\Delta(R\text{-}F)$, thereby leading to spread in the estimated $\lambda$ among
the ensemble members.  We also see strong variability in $\lambda_r$ within each region, suggesting that
how the warming is distributed within the region also drives some of the spread in estimated $\lambda$
in the ensemble.
This explanation is consistent with analyses showing that $\lambda$ changes during transient runs as the
pattern of surface temperature evolves (Senior and Mitchell, 2000; Armour et al., 2013;
Andrews et al., 2015; Gregory and Andrews, 2016; Stevens et al., 2016).  In our model
ensemble, however, the pattern changes are caused by internal variability rather than differing
regional heat capacities that cause some regions to warm more slowly than others during
forced warming.
**A better way to describe energy balance**
Our analysis demonstrates limitations of the conventional energy balance framework (Eq. 1). It
has been previously noted that $\Delta R$ correlates better with tropospheric temperatures than $\Delta T_S$
(Murphy, 2010; Spencer and Braswell, 2010; Trenberth et al., 2015). Recent analyses have also
stressed the importance of atmospheric temperatures — through its influence on lapse rate —
as providing a fundamental control on the planet's energy budget (Zhou et al., 2016; Ceppi and
Gregory, 2017).  Based on this, we test a new energy balance framework constructed using the
temperature of the tropical atmosphere:





$R - F = \Theta\, T_A$      (4)
where $T_A$ is the tropical average (30°N-30°S) 500-hPa temperature and $\Theta$ relates this quantity to
R-F. R and F are the same global average quantities they were in equation 1. ECS can be
expressed in terms of $\Theta$:
$$ECS = -\frac{\Delta F_{2\times CO2}}{\Theta}\frac{\Delta T_S}{\Delta T_A}$$      (5)
where $\Delta T_S$ and $\Delta T_A$ are the equilibrium changes in these quantities in response to doubled $CO_2$;
the CMIP5 ensemble average ratio $\Delta T_S/\Delta T_A$ is 0.86±0.10 (±1σ), where $\Delta$ represents the average
difference between the first and last decades of the abrupt $4xCO_2$ runs.
Support for Eq. 4 can be found in the observations: $\Delta R$ shows a tighter correlation with $\Delta T_A$ than
with $\Delta T_S$ in observations (Figs. 3a vs. 3b). Given that the slope of these plots can be taken as
estimates of $\Theta$ and $\lambda$, the tighter correlation leads to more accurate estimates of $\Theta$ than $\lambda$,
both in absolute and relative terms.
Turning to the model ensemble, we next demonstrate that $\Theta$ is a more precise metric than $\lambda$.
We do this by calculating $\Theta$ [= $\Delta$(R-F)/$\Delta T_A$] in each ensemble member, yielding values ranging
from -1.18 to -0.89 W/m$^2$/K, with an ensemble average of -1.04 W/m$^2$/K (Fig. 2a). There is
clearly less variability in $\Theta$ among the ensemble members than for $\lambda$. This reflects less
variability in the regional response $\Theta_r$ (= $\Delta$(R-F)$_r$/$\Delta T_{A,r}$) than $\lambda_r$ (Fig. 5), as well as less variability
within the regions. We therefore conclude that interannual variability has less of an impact on
$\Theta$ than $\lambda$. We show additional evidence for the superior precision of $\Theta$ in the Appendix.
As far as accuracy goes, we can compare $\Theta$ in the ensemble over the historical period to $\Theta$ in
response to much larger warming. The ensemble average $\Theta$ from the historic period, -1.04
W/m$^2$/K, is close to the value obtained from an analysis of the first 150 years of an abrupt
$4xCO_2$ run of the same model, $\Theta$ = -1.03 W/m$^2$/K, as well as $\Theta$ calculated from all 2600 years of
this run, $\Theta$ = -1.04 W/m$^2$/K. On the other hand, $\lambda$ changes substantially in the $4xCO_2$ run as the
climate warms: $\lambda$ = -1.36 W/m$^2$/K when calculated from the first 150 years, but $\lambda$ = -0.95
W/m$^2$/K from all 2600 years of that run.



We can verify this result in the CMIP5 abrupt $4xCO_2$ ensemble. It has been previously
demonstrated that plots of R-F vs. $T_S$ do not trace straight lines as the climate warms (Andrews
et al., 2015; Rugenstein et al., 2016; Rose and Rayborn, 2016; Armour, 2017), so $\lambda$ and ECS
calculated in a single model run may depend on the portion of the run selected. In the CMIP5
abrupt $4xCO_2$ ensemble, for example, average $\lambda$ calculated by regressing years 10-30 ($\lambda_{10-30}$) is
more negative than $\lambda$ calculated from years 30-150 ($\lambda_{30-150}$) by 0.50 $W/m^2/K$ (Fig. 6).
Several explanations for this have been advanced, most prominently that $\lambda$ is function of the
pattern of surface warming (Senior and Mitchell, 2000; Armour et al., 2013; Andrews et al.,
2015; Gregory and Andrews, 2016; Zhou et al., 2016; Stevens et al., 2016). Using $\Theta$ largely
eliminates this pattern effect: $\Theta_{10-30}$ and $\Theta_{30-150}$ have an average difference of 0.16 $W/m^2/K$ for
the CMIP5 ensemble (Fig. 6). Thus, we find additional evidence that $\Theta$ tends to be similar for
different amounts and patterns of warming.
Finally, one of our ultimate goals for this revised framework is to help produce better estimates
of ECS. We are working on a detailed analysis of ECS based on this framework and will publish
that in a follow-on paper, but we briefly show here how the advantages of the revised energy
balance framework may be leveraged to do this. Fig. 7a shows $\Theta$ calculated from control runs
of 25 CMIP5 models. To calculate $\Theta$ in the control runs, we break each control run into 16-year
segments and calculate monthly anomalies of $\Delta R$ and $\Delta T_A$ during each segment. Then, we
calculate $\Theta$ for each segment as the slope of the regression of $\Delta R$ vs. $\Delta T_A$ for that segment.
Thus, for each control run, we generate a large number of estimates of $\Theta$. The value in Fig. 7a is
the average of these individual values.
Fig. 7b shows the ECS of these models, calculated from the first 150 years of the abrupt $4xCO_2$
runs using the Gregory method (Gregory et al., 2004). If we assume that models with more
accurate simulation of short-term $\Theta$ produce more accurate estimates of ECS (Brown and
Caldeira, 2017; Wu and North, 2002), then we can use Figs. 7a and 7b to constrain ECS. We find
that the 15 models whose short-term $\Theta$ agrees with the CERES observations have ECS values
ranging from 2.0-3.9 K, with an average of 2.9 K. This excludes many of the highest ECS models.





It would not have been possible to draw this conclusion with the conventional energy balance
framework. Fig. 7c shows the comparison between λ from the control runs (calculated the
same way Θ was calculated) and CERES observations.  Because of the much larger uncertainty
in the observational estimate of short-term λ, almost all models fall within the observational
range, thereby prohibiting any constraint on the ECS range.
**Conclusions**
We have estimated ECS in each of a 100-member climate model ensemble using the same
energy-balance constraint used by many investigators to estimate ECS from 20[th]-century
historical observations.  We find that the method is imprecise — the estimates of ECS range
from 2.1 to 3.9 K (Fig. 2), with some ensemble members far from the model's true value of 2.9
K.  Given that we only have a single ensemble of reality, this suggests that some skepticism is
appropriate when considering estimates of ECS derived from the historical record.
The source of the imprecision relates to the construction of the traditional energy balance
equation (Eq. 1).  In it, the response of TOA net flux (R-F) is parameterized in terms of global
average surface temperature ($T_S$).  Recent research has suggested that the response is not just
determined by the magnitude of $T_S$, but includes other factors, such as the pattern of $T_S$ (e.g.,
Armour et al., 2013; Andrews et al., 2015; Gregory and Andrews, 2016; Zhou et al., 2017) or the
lapse rate (e.g., Zhou et al., 2017; Ceppi and Gregory, 2017).  As a result, two ensemble
members with the same $\Delta T_S$ can have different climate responses, $\Delta(R\text{-}F)$, leading to spread in
the inferred λ.
The lack of a direct relationship between $T_S$ and radiation balance suggests that it may be
profitable to investigate alternative formulations. We test parameterizing the response in terms
of 500-hPa tropical temperature (Eq. 4) and find that it is superior in many ways.  Ultimately,
how investigators describe the energy balance of the planet will depend on the problem and
the available data.  The surface temperature is indeed special, so the traditional framework
may be preferred for some problems.  But investigators may find that the alternatives are
superior for certain problems, for instance constraining Earth's climate sensitivity.





Appendix
It has been previously noted in analyses of the historical record that λ exhibits significant
interdecadal variability (Andrews et al., 2015; Gregory and Andrews, 2016; Zhou et al., 2016).
We can reproduce this in a 2000-year control run (a run with fixed pre-industrial boundary
conditions) of the MPI-ESM1.1 model.  Fig. 8 shows λ calculated in a sliding 16-year window
and confirms significant temporal variability in λ.  We can similarly calculate Θ and find that
temporal variability in Θ is substantially smaller (Fig. 8).
This result is reproduced in the CMIP5 control models.  Fig. 9 plots the standard deviation of
each CMIP5 model's set of short-term λ divided by the standard deviation of that model's set of
short-term Θ (as described previously, we calculate time series of short-term λ and Θ values for
each model by regressing anomalies in a 16-year sliding window of the control runs).  All of the
models fall above 1, demonstrating that there is less variability in the Θ time series than in the
λ time series in every climate model.  This confirms that Θ is more robust with respect to
internal variability than λ. It also suggests that Θ estimated from the satellite data (Fig. 3)
should be considered a better estimate of the climate system's long-term value than λ
estimated from the same data set.



Acknowledgements: This work was supported by NSF grant AGS-1661861 to Texas A&M
University.  This work was completed while AED was on Faculty Development Leave from Texas
A&M during the Fall of 2016; he thanks Texas A&M and the Max Planck Institut für
Meteorologie for supporting this research.  Computational resources were made available by
Deutsches Klimarechenzentrum (DKRZ) through support from German Federal Ministry of
Education and Research (BMBF), and by the Swiss National Supercomputing Centre (CSCS).
CERES data were downloaded from ceres.larc.nasa.gov, ECMWF-interim data were downloaded
from http://apps.ecmwf.int/datasets/.



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



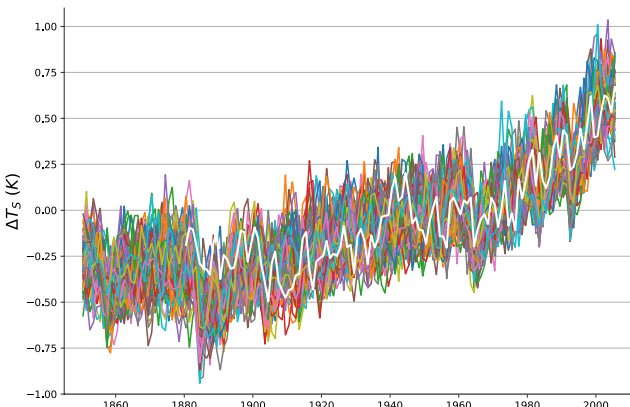


Fig. 1. Plot of annual and global average surface temperature from the 100 members of the
MPI-ESM1.1 ensemble (colored lines), along with the GISTEMP measurements (Hansen et
al., 2010) (white line).  Temperatures are referenced to the 1951-1980 average.

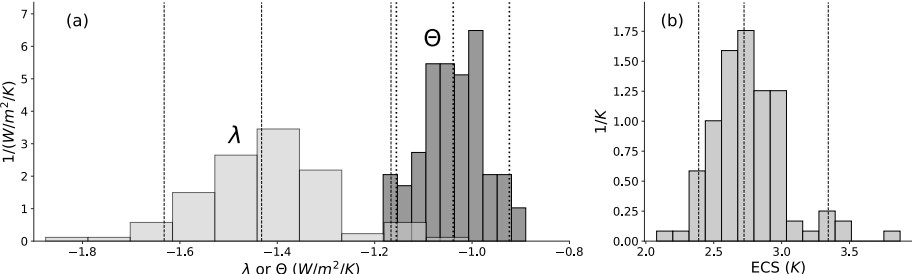


Figure 2. PDFs of (a) λ (lighter) and Θ (darker) and (b) ECS derived from the members of the
MPI-ESM1.1 historical ensemble. The vertical lines are the 5[th], 50[th], and 95[th] percentile of each
distribution.



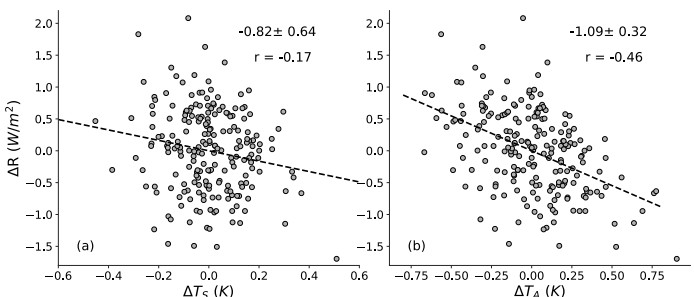


Figure 3. Scatter plot of detrended monthly anomalies of ΔR vs. (a) global average surface
temperature $\Delta T_S$, (b) tropical average 500-hPa temperature $\Delta T_A$. Observations cover the period
March 2000-Jan. 2017 and anomalies are deviations from the mean annual cycle.  The dashed
lines are ordinary least-squares fits; the slope, 5-95% confidence interval, and correlation
coefficient are shown on each panel.  Confidence intervals account for autocorrelation of the
time series (Santer et al., 2000).



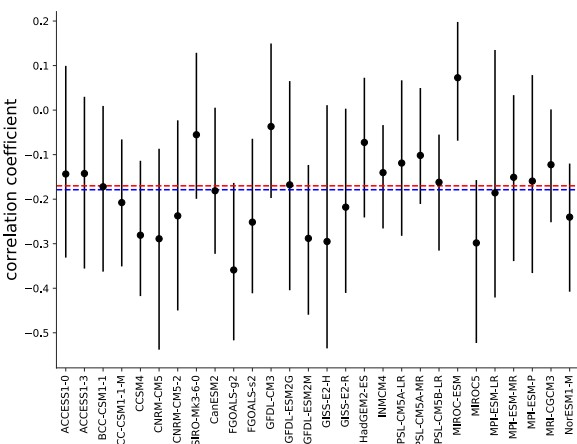


Fig. 4. Correlation coefficients between ΔR and ΔT$_S$ in CMIP5 control runs.  The dot is the

average of the correlation coefficients from the 16-year segments of the model run; the

bars indicate the maximum and minimum values from the control run.  The blue dashed

line is the average of the CMIP5 models, while the red dashed line is the correlation

coefficient from the CERES regression in Fig. 2a.

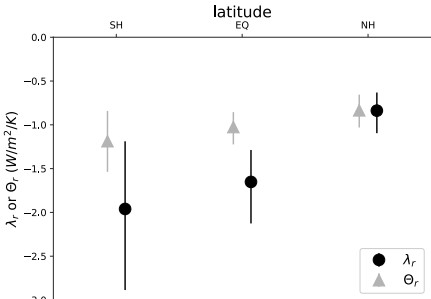

386

Fig. 5. λ$_r$ and Θ$_r$ calculated as regional average Δ(R-F) divided by regional average temperature

(ΔT$_S$ for λ and ΔT$_A$ for Θ).  The regions are 90°S-19.4°S (SH), 19.4°S-19.4°N (EQ), and 19.4°N-

90°N (NH).  The values are calculated for each member of the 100-member ensemble; the solid

symbols are the ensemble average while the bars show the 5-95% range.

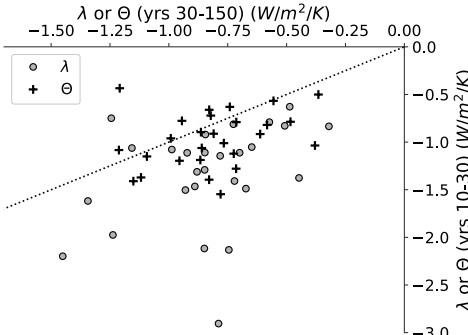

391

Fig. 6. Scatterplot of $\lambda_{10-30}$ vs. $\lambda_{30-150}$ (gray circles) in CMIP5 abrupt4xCO₂ runs, as well as

$\Theta_{10-30}$ vs. $\Theta_{30-150}$ (black crosses) in the same models. Each point represents one model.

The dotted line is the 1:1 line. The subscripts (10-30, 30-150) indicate the years of the run

from which the values are calculated.

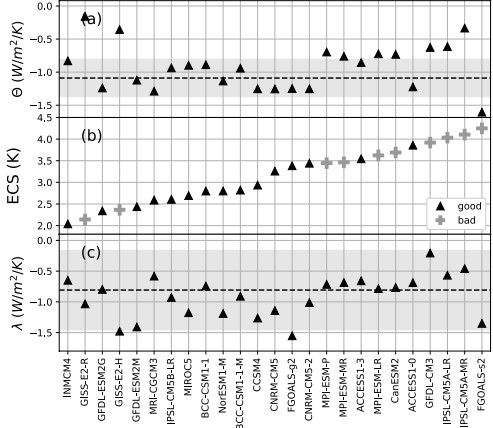

396

Figure 7. (a) $\Theta$ from individual CMIP5 control runs (calculation described in the Appendix).

The dotted line is the estimate from CERES observations; the gray region is the 5-95%

confidence band. (b) ECS from each CMIP5 model, estimated from the first 150 years of

abrupt 4xCO₂ runs using the Gregory method (Gregory et al., 2004). "Good" models are

those whose $\Theta$ agrees with observations, "bad" models are those that do not. (c) Same as

panel (a), but for $\lambda$.






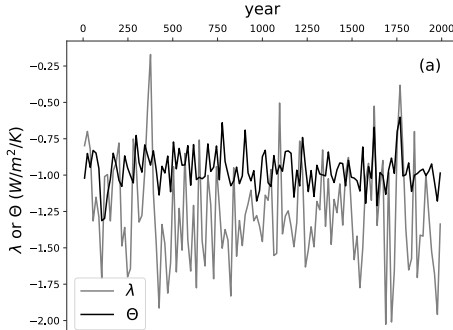

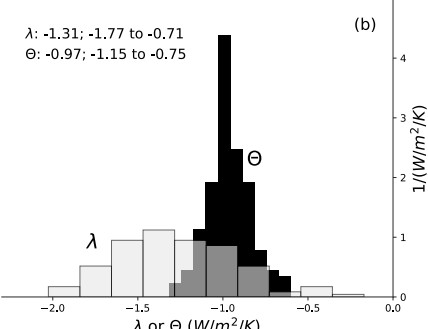


Fig. 8. (a) Time series of λ (gray) and Θ (black) estimated in a 16-year sliding window of a
2000-year control run of the MPI-ESM1.1. (b) PDFs of the time series in panel a. Median
and 5-95% confidence interval for each distribution is displayed on the plot.





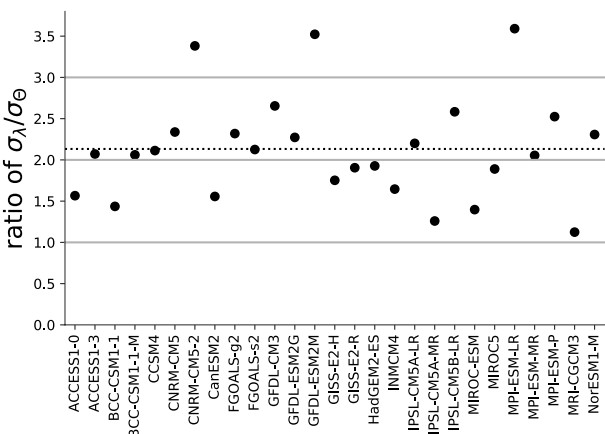


Fig. 9. The standard deviation of the λ time series divided by the standard deviation of the

Θ time series. Each time series is calculated from 16-year segments of CMIP5 control runs.

The dotted line is the ensemble average.
