# Peer review of "The influence of internal variability on Earth's energy balance framework and implications for"

_Atmospheric Chemistry and Physics, 2017_

## Short Comment (SC1) · 17 Jan 2018

It seems to me the key underlying assumption in the paper is that the specific model used (MPI-ESM1.1) has internal variability which is an accurate representation of the Earth's internal variability. I don't think the authors have shown this is true. That is, if the model's surface temperature is considerably more variable than Earth's actual surface temperature history, that would suggest less correlation in the model between a change in surface temperature and a change in loss of heat to space than is correct. The spaghetti graph in the paper, which overlays 100 model runs (100 runs!?!) and compares to the GISS history, obscures how much internal variability there is in the

individual runs.

I find the arguments about modeled temperature changes at 500 mb unconvincing. That is not how EB estimates of ECS have been done, and in any case, it seems irrelevant to the paper's central claim that Earth's surface temperature has too much internal variability to generate a useful estimate of climate sensitivity.

I have never looked specifically at individual runs of this model, but I have looked at individual runs from several other models, and many consistently display much more short term variability than the instrumental temperature history shows. This did not surprise me at all, since models which are too sensitive to forcing are likely going to display higher short term variability.

The paper could be improved by comparing the GISS and Hadley temperature histories to a dozen or two randomly selected individual model runs, on 4 or 6 graphs, so that any differences variability could be visually compared. We should expect to see at least some runs where model variability is comparable to or less than measured variability. If all model runs are more variable than the historical record, I think that cases serious doubt on the accuracy of the key underlying assumption. The paper could be improved much more by calculating the variability in surface temperature for each modeled run as the total range in temperature anomaly over a few different time windows; eg. total temperature range over 5 year, 10 year, and 20 year rolling boxcar periods, and comparing to the same range values from the temperature history. If the model is a reasonable representation of Earth's internal variability, then the variability for the temperature history will fall well within the distribution of variability for the individual model runs.

---

## Short Comment (SC2) · 17 Jan 2018

The main question in my mind is whether GCM's are skillful for the kind of tropical temperatures (500 hPa) being used here to correlate with energy imbalances.

A recent paper (Zhoa et al 2016) shows that tropical convection sub grid model parameters have a strong influence on model ECS and that there are no obvious constraints to set those parameters. Another shows that diurnal cloudiness errors in models have a significant influence on insolation at the surface. There are also recent papers showing that adding convective aggregation to a model significantly reduces ECS. And yet another showing that some of these changes affect the vertical temperature gradient

in the tropics.

GCM's are rather good at predicting Rossby waves. It would seem however that other measures of skill, particularly in the tropics are not as firmly established.
* * *

---

## Author Comment (AC1) · 19 Jan 2018

The author of this comment questions our study because they do not have confidence that the relationships which we rely on are well captured by the models. Our findings are of course conditioned on the idea that the model can usefully calculate the TOA energy balance given the state of the climate system (i.e., the distribution of temperature and water vapor, etc.). The author gives several reasons why the class of comprehensive models that we use are inadequate in one respect or another, and we share their frustration regarding model failings. However none of the failings they point out, or that we are aware of, seem relevant to the key hypothesis that we use the models to help

quantify, namely that TOA flux is more tightly tied to atmospheric temperatures than to surface temperatures. We note that this conclusion is also supported by observations, and is sensible even without having looked at a model because surface temperatures can decouple from the radiation balance as a whole quite easily, i.e., land surface temperature in winter, see e.g., (Li, Marotzke, and Stevens, 10.1002/2015GL065327, 2016).

---

## Author Comment (AC2) · 19 Jan 2018

Comment: It seems to me the key underlying assumption in the paper is that the specific model used (MPI-ESM1.1) has internal variability which is an accurate representation of the Earth's internal variability. I don't think the authors have shown this is true. That is, if the model's surface temperature is considerably more variable than Earth's actual surface temperature history, that would suggest less correlation in the model between a change in surface temperature and a change in loss of heat to space than is correct.

Response: Ours is a methodological study that asks the question: does the method

used to obtain ECS from a single realization of the historical record warming work in a perfect model experiment? We find the answer is that there is considerable uncertainty associated with single realization estimates, as show in Fig. 2. This is the result of internal variability in the model. While it is difficult to rigorously validate internal variability in models in general, much work on this has been done (see, e.g., Sect. 9.5 of the IPCC Fifth Assessment Report, Working Group 1) and the models' variability seems reasonable. For this specific model, there was also an evaluation done for variability in the global radiation budget (Hedemann et al. 2017, Fig. S7). That said, we stress that none of this matters for the purpose of this perfect-model study.

Comment: The spaghetti graph in the paper, which overlays 100 model runs (100 runs!?!) and compares to the GISS history, obscures how much internal variability there is in the individual runs.

Response: There seems to be some confusion about what we mean when we talk about "internal variability". Internal variability in this paper refers to variability in the pattern of surface warming, not variability in the global average warning. In other words, given an amount of global warming, internal variability is a measure of the different ways the warming can be distributed around the globe, which in turn affects the planetary radiation balance. This is what interferes with estimates of ECS from the historical record.

Comment: I find the arguments about modeled temperature changes at 500 mb unconvincing. That is not how EB estimates of ECS have been done, and in any case, it seems irrelevant to the paper's central claim that Earth's surface temperature has too much internal variability to generate a useful estimate of climate sensitivity.

Response: Our discussion of 500-hPa temperature is presented as a different and in many ways superior description of the energy balance of the planet. It has little bearing on the argument that using surface temperature produces an imprecise estimate of ECS.

Comment: I have never looked specifically at individual runs of this model, but I have looked at individual runs from several other models, and many consistently display much more short term variability than the instrumental temperature history shows. This did not surprise me at all, since models which are too sensitive to forcing are likely going to display higher short term variability. The paper could be improved by comparing the GISS and Hadley temperature histories to a dozen or two randomly selected individual model runs, on 4 or 6 graphs, so that any differences variability could be visually compared. We should expect to see at least some runs where model variability is comparable to or less than measured variability. If all model runs are more variable than the historical record, I think that cases serious doubt on the accuracy of the key underlying assumption. The paper could be improved much more by calculating the variability in surface temperature for each modeled run as the total range in temperature anomaly over a few different time windows; eg. total temperature range over 5 year, 10 year, and 20 year rolling boxcar periods, and comparing to the same range values from the temperature history. If the model is a reasonable representation of Earth's internal variability, then the variability for the temperature history will fall well within the distribution of variability for the individual model runs.

Response: This emphasis on global average variability is not what we mean when we talk about internal variability. See comment above.

Hedemann, C. et al. The subtle origins of surface-warming hiatuses. Nature Climate Change, doi:10.1038/nclimate3274, 2017.

---

## Short Comment (SC3) · 2 Feb 2018

In this paper, the authors present an interesting result. Based on both historical observations and analyses of results from an advanced climate model they found that $\Delta(R - F)$ correlates much better with changes in the mean tropical tropospheric temperature $\Delta T_A$ than with changes in the mean global surface temperature $\Delta T_S$. Thus, the linearised equation for the TOA radiative flux imbalance

$$R = F + \Theta T_A \tag{1}$$

gives a more precise representation of $R - F$ than the traditional linearised equation

that has frequently been used for determining ECS from historical observations

$$R = F + \lambda T_S \qquad (2)$$

Consequently, the climate feedback parameter $\Theta$ can be determined more precisely than the traditional climate feedback parameter $\lambda$.

However, this paper also raises a question of a fundamental character concerning how regional versions of Eqs. (1) and (2) relate to the original global ones. The authors have discussed regional versions of Eqs. (1) and (2) in the paper

$$R_r = F_r + \lambda_r T_{S,r} \qquad (3)$$
$$R_r = F_r + \Theta_r T_{A,r} \qquad (4)$$

and found from the observations of the TOA imbalance that the three different regions NH, EQ and SH exhibited different values of $\lambda_r$ and $\Theta_r$ (see Fig. 5 in the paper). This raises a fundamental question about the compatibility of Eqs. (2) and (3) or (1) and (4) that may be illustrated by an example with a planet being described using three regions with the surface area fractions $a_1$, $a_2$ and $a_3$. By using the three equations corresponding to Eq. (3) and applying $R = a_1 R_1 + a_2 R_2 + a_3 R_3$, $F = a_1 F_1 + a_2 F_2 + a_3 F_3$, $\lambda = a_1 \lambda_1 + a_2 \lambda_2 + a_3 \lambda_3$ and $T_S = a_1 T_{S,1} + a_2 T_{S,2} + a_3 T_{S,3}$ we get the following equivalent expressions for the TOA radiative flux imbalance

$$R = F + \lambda_1 T_S + a_2(\lambda_2 - \lambda_1)T_{S,2} + a_3(\lambda_3 - \lambda_1)T_{S,3} \qquad (5)$$
$$R = F + \lambda_2 T_S + a_1(\lambda_1 - \lambda_2)T_{S,1} + a_3(\lambda_3 - \lambda_2)T_{S,3} \qquad (6)$$
$$R = F + \lambda_3 T_S + a_1(\lambda_1 - \lambda_3)T_{S,1} + a_2(\lambda_2 - \lambda_3)T_{S,3} \qquad (7)$$
$$R = F + \lambda T_S + a_2 a_1(\lambda_2 - \lambda_1)(T_{S,2} - T_{S,1}) +$$
$$+ a_3 a_1(\lambda_3 - \lambda_1)(T_{S,3} - T_{S,1}) + a_3 a_2(\lambda_3 - \lambda_2)(T_{S,3} - T_{S,2}) \qquad (8)$$

Similar equations are obtained with $T_A$ instead of $T_S$ and $\Theta$ instead of $\lambda$. The result of this analysis is that the linearised regional TOA imbalance equations are not in general

compatible with the linearised global TOA imbalance equation. If Eq. (3) is satisfied for each region, then Eq. (2) is in general not satisfied globally. While Eq. (2) expresses $R - F$ as a function of $T_S$ alone, according to Eqs. (5)–(8) $R - F$ in general is a function three temperatures.

The authors have demonstrated the estimation of the regional climate feedback parameters, both $\lambda_r$ and $\Theta_r$, from Eqs. (3) and (4). The values of those parameters given in Fig. 5 in their paper show that the differences between the $\Theta_r$-values are less than the differences between the $\lambda_r$-values. From the form of Eq. (8) this suggests that Eq. (1) should give a better correlation than Eq. (2), in agreement with the author's results.

In the example given here regional ECSs could be calculated as $\text{ECS}_r = \Delta T_{S,r} = -\Delta F_{2 \times CO2,r}/\lambda_r$. Then, the global ECS simply equals $\text{ECS} = a_1 \Delta T_{S,1} + a_2 \Delta T_{S,2} + a_3 \Delta T_{S,3}$. Perhaps informed choices of regions could produce regional climate feedback parameters that allow ECS values more adapted to the effects of warming patterns and internal variability? Perhaps it is better to characterize the sensitivity of the climate for radiative forcing by several regional climate feedback parameters and regional ECS values, instead of only by one global climate feedback parameter and by one global ECS?

---

## Author Comment (AC3) · 2 Feb 2018

We agree entirely with this comment. Our revised energy balance framework (Eq. 4 in our paper) is a "proof of concept" that demonstrates that it is possible to do a better job describing Earth's energy balance than the conventional approach does. However, we don't expect it to be the final answer and agree with the commenter that a version using several regional temperatures may be superior.

---

## Referee Comment (RC1) · Anonymous Referee #1 · 6 Feb 2018

The main point of this paper is that the radiative response at the TOA to an imposed radiative forcing relates more directly to, and correlates better with, the temperature change in the tropical free troposphere than it does with the global mean surface air temperature change. They support this with evidence from CMIP5 AOGCMs and the large ensemble of the MPI-ESM1.1 AOGCM. They further argue that the tropospheric temperature is therefore better than the surface temperature for use in the Earth energy balance framework, and show that climate feedback defined in those terms is more constant. This is a reasonable point. However, surface temperature may relate more to some climate feedbacks and to impacts of climate change. Hence their reformulation of the energy balance may to some extent shift the problem elsewhere, because the

relationship between mean tropospheric temperature and the surface temperature and its patterns still has to be separately understood. They acknowledge this limitation at the end of the Conclusions. I think the statement at line 232 could be qualified as well, because this depends on the length of record being used, surface temperature having longer global records.

The paper is clearly written and illustrated. I have some minor points for the authors to consider.

64-65. As a single number to quantify the spread, the standard deviation would also be helpful.

66. Why do you use only a single decade, rather than all the data, for instance by dividing the dataset into two or using regression (cf Barnes and Barnes, 2015, 10.1175/JCLI-D-15-0032.1)? A single decade would be less precise. You could estimate the statistical uncertainty incurred from the control run.

118. It would be useful to remark here that 16 years is chosen to match the CERES dataset, because that was mentioned some lines above (103-104), where it appears actually to be 17 years and 5 months long.

119, 196. Why are monthly anomalies used here, rather than annual? Does it make a difference?

167. Again, the standard deviation would be helpful, and could be compared with lines 64-65.

173, 175. You could give standard errors of the mean for each of these two numbers, and judge the significance of their difference.

174, 175. "analysis" and "calculated" - by what method? From the slope of R against Delta T?

204. "agrees" in what sense?

218. I would say that this is "one source" of the spread, which is not eliminated, but only reduced, by using Theta instead.

233. Why is this material an appendix, rather than being incorporated in the main text?

---

## Referee Comment (RC2) · Anonymous Referee #2 · 9 Feb 2018

Summary: This paper shows that the traditional energy balance framework yields a poor representation of the Earth's energy budget when unforced variability is large, because the relationship between radiative response and global-mean surface temperature is weak. The authors then propose an improved energy balance relationship where the radiative response is assumed to scale with tropical-mean 500 hPa temperature. Using a large ensemble of historical experiments, the authors demonstrate that this relationship yields more accurate estimates of the magnitude of climate feedbacks under CO2 forcing.

Recommendation: Minor revision

[Figure]

Comments: I enjoyed reading this paper, which is clear and concise. The idea is interesting and for the most part I am convinced by the arguments presented by the authors. However, I would like them to show some additional evidence, as described below, and to better discuss some of the potential limitations of the proposed energy balance relationship.

1) It would be helpful to provide a little more physical motivation for the choice of tropical 500 hPa temperature. I see some good reasons why mid-tropospheric temperature should work better (e.g., it should scale better with LR, WV and LW cloud feedbacks), but I don't think this was discussed anywhere. Why use tropical temperature rather than global-mean? Is there a physical rationale, or did this simply work better in MPI-ESM?

Also, although mid-tropospheric temperature clearly works better for the overall feedback, I expect the scaling with Ta might actually be a worse choice for some individual feedback processes (e.g. surface albedo, marine low cloud). This might be worth discussing briefly.

2) A key result is that the revised feedback parameter $\theta$ more accurately estimates the "true" feedback strength under CO2 forcing. This is shown to be the case in MPI-ESM (L172-176). However, does this hold for CMIP5 models in general? I.e., do the values of $\theta$ estimated in control runs correlate well with those in 4xCO2?

Relatedly, I would also suggest adding the correlation between $\Delta R$ and $\Delta Ta$ in CMIP5 piControl to Fig. 4, as additional bars in a different color.

3) One important issue that isn't discussed in the paper is that the "pattern effect" doesn't simply go away with the improved relationship; rather, it shifts from the feedback parameter to the $\Delta Ts/\Delta Ta$ term. This isn't a problem, but the way the paper is currently written, some readers might get that impression.

So if most of the curvature in the relationship between radiative response and temperature goes away with the revised framework (Fig. 6), I expect there must be some

curvature in the Ta versus Ts relationship in 4xCO2 runs. Can the authors confirm this?

4) I expect the $\Delta$Ts/$\Delta$Ta ratio cannot be reliably estimated from historical runs in the presence of large variability (for the same reason that $\lambda$ cannot be reliably estimated - because of the pattern effect). So we must rely on models to estimate this ratio under future global warming, meaning that it will be important to understand how future patterns of surface warming will develop. I suggest the authors discuss this briefly, for example in the conclusions.

Other minor comments: - I suggest using colors in Fig. 6, rather than dark grey and black. - L223: Cite Andrews and Webb 2018 - For future reference, it would be useful to mention the value of $\theta$ estimated from observations (horizontal dashed bar in Fig. 7a).

---

## Short Comment (SC4) · 11 Feb 2018

Like this reviewer, I enjoyed reading the authors' paper on interesting fundamental issues concerning the traditional energy balance framework. The reviewer suggests some further justification of why the tropical 500 hPa temperature should work better than the global mean. I have some thoughts on this as an extension to my previous short comment SC3.

In the discussion paper the climate feedback parameter $\Theta = \Delta(R - F)/\Delta T_A$ with reference to the mean 500 hPa tropical temperature anomaly $T_A$, where the tropics are defined as 30°N to 30°S, is discussed. There are also three regional feedback

parameters considered $\Theta_r = \Delta(R - F)_r/\Delta T_{A,r}, r = 1, 2, 3$, where index $r$ denotes the three regions 90°N to 19.4°N, 19.4°N to 19.4°S and 19.4°S to 90°S (see Fig. 5 in the discussion paper). However, by definition $T_{A,r}$ denotes the corresponding temperature anomalies averaged over each of the three tropical regions 30°N to 19.4°N, 19.4°N to 19.4°S and 19.4°S to 30°S.

As discussed in my previous comment SC3, if the regional parameters would be constant the global parameter will in general not be a constant but instead satisfying the following equation

$$
\begin{aligned}
R \; = \; & F + \Theta T_A + a_2 a_1 (\Theta_2 - \Theta_1)(T_{A,2} - T_{A,1}) + \\
& + a_3 a_1 (\Theta_3 - \Theta_1)(T_{A,3} - T_{A,1}) + a_3 a_2 (\Theta_3 - \Theta_2)(T_{A,3} - T_{A,2})
\end{aligned} \tag{1}
$$

where $R$ is the global TOA flux imbalance and F is the global radiative forcing.

The tropics have less seasonal and regional temperature variations than other regions on the planet, for example, the subarctic one. Thus, the differences in temperatures $T_{A,r}$ between the three tropical regions can be expected to be less than between the three whole regions, covering the whole planet. Consequently, it may be expected that the three last terms in Eq. (1) are adding less flux variation than the three last terms in Eq. (8) in SC3. This may also be valid in case of a corresponding equation using the 500 hPa temperatures, averaged not only over the tropical parts but over the whole regions. This is one reason why a better correlation between $\Delta(R - F)$ and $\Delta T_A$ than between $\Delta(R - F)$ and some form of global mean temperature may be expected.

Furthermore, the differences between the feedback parameters $\Theta_r$ were less than between the parameters $\lambda_r$ based on the global mean surface temperature (see Fig. 5 in the discussion paper). This gives also a contribution to the better correlation as discussed in SC3. Perhaps this less difference between the $\Theta_r$ parameters may be explained by the changes in the TOA flux being dominated by the changes of the LW radiation and the tropics having a dominating role for the changes of the LW radiation?

---

## Short Comment (SC5) · 13 Feb 2018

Equation (1) in my comment SC4 has the same shape as Eq. (8) with respect to the traditional climate feedback parameter $\lambda$ displayed in my comment SC3

$$
\begin{aligned}
R &= F + \lambda T_S + a_2 a_1 (\lambda_2 - \lambda_1)(T_{S,2} - T_{S,1}) + \\
&\quad + a_3 a_1 (\lambda_3 - \lambda_1)(T_{S,3} - T_{S,1}) + a_3 a_2 (\lambda_3 - \lambda_2)(T_{S,3} - T_{S,2})
\end{aligned} \tag{1}
$$

where it is implied that $R = a_1 R_1 + a_2 R_2 + a_3 R_3$, $F = a_1 F_1 + a_2 F_2 + a_3 F_3$, $\lambda = a_1 \lambda_1 + a_2 \lambda_2 + a_3 \lambda_3$ and $T_S = a_1 T_{S,1} + a_2 T_{S,2} + a_3 T_{S,3}$. Here $a_1$, $a_2$ and $a_3$ are the area fractions of the three regions 90°N to 19.4°N, 19.4°N to 19.4°S and 19.4°S to 90°S.

[Figure]

However, $T_S$ is a global average. In the case with the climate feedback parameter $\Theta$ introduced in the discussion paper, the 500 hPa mean tropical temperature $T_A$ is used instead of $T_S$. Because $T_A$ is averaged over the tropical regions 30°N to 19.4°N, 19.4°N to 19.4°S and 19.4°S to 30°S there are different area fractions for $T_A$ than for $R$, $F$ and $\Theta$. This was overlooked in SC4. The correct equation with $T_A = b_1 T_{A,1} + b_2 T_{A,2} + b_3 T_{A,3}$ should be:

$$
\begin{aligned}
R &= F + \Theta T_A + (a_2 b_1 \Theta_2 - a_1 b_2 \Theta_1)(T_{A,2} - T_{A,1}) + \\
&\quad + (a_3 b_1 \Theta_3 - a_1 b_3 \Theta_1)(T_{A,3} - T_{A,1}) + (a_3 b_2 \Theta_3 - a_2 b_3 \Theta_2)(T_{A,3} - T_{A,2}) \quad (2)
\end{aligned}
$$

───────────────────

---

## Author Comment (AC4) · 16 Mar 2018

We thank the reviewer for their comments. In this document, we detail our responses.

64-65. As a single number to quantify the spread, the standard deviation would also be helpful.

A: We have added 5-95% confidence intervals throughout the paper.

66. Why do you use only a single decade, rather than all the data, for instance by dividing the dataset into two or using regression (cf Barnes and Barnes, 2015, 10.1175/JCLI-D-15-0032.1)? A single decade would be less precise. You could es-

timate the statistical uncertainty incurred from the control run.

A: We calculate ECS using this approach because this is the way most ECS calculations based on the 20th-century observational record are done. Thus, our results can therefore directly provide insight into the impact of variability in the observational estimates of ECS.

The reviewer is correct that using more than a decade might affect the results. If one used the difference between the averages of the first and last 20 years, the range in lambda declines from 0.87 W/m2/K to 0.48 W/m2/K. Using longer averaging periods does not further decrease the range. We now mention this in the paper.

118. It would be useful to remark here that 16 years is chosen to match the CERES dataset, because that was mentioned some lines above (103-104), where it appears actually to be 17 years and 5 months long.

A: We have added a statement that the segmentation of the data is done to match the CERES record. We have also updated the paper to segment the data into 17-year segments to more closely match CERES.

119, 196. Why are monthly anomalies used here, rather than annual? Does it make a difference?

A: We do this to facilitate the comparison with the CERES regressions, which also uses monthly data. The reason most analyses with CERES data are done with monthly data is because using annual data means there's only 17 data points, and the uncertainties end up being very large. Issues involved in annual vs. monthly regressions are discussed in some detail in Forster (2016, 10.1146/annurev-earth-060614-105156).

167. Again, the standard deviation would be helpful, and could be compared with lines 64-65.

A: Added.

173, 175. You could give standard errors of the mean for each of these two numbers, and judge the significance of their difference.

A: We have added the 5-95% confidence intervals to all of these numbers.

174, 175. "analysis" and "calculated" - by what method? From the slope of R against Delta T?

A: We have clarified the text that we use the method of Gregory et al. (2004), where annual average R is regressed against T, and the slope of the curve is an estimate of lambda or theta.

204. "agrees" in what sense?

A: We have changed the sentence to read: "We find that the 15 models whose short-term theta falls within the uncertainty of theta estimated from CERES observations have ECS values ranging from 2.0-3.9 K, with an average of 2.9 K."

218. I would say that this is "one source" of the spread, which is not eliminated, but only reduced, by using Theta instead.

A: We would argue that this sentence is phrased correctly. The spread in our estimate from the ensemble is due to the construction of the energy-balance equation. Unlike observational analyses, we know everything else perfectly. Using our revised energy balance equation does not completely solve the problem, but it is an improvement.

233. Why is this material an appendix, rather than being incorporated in the main text?

A: We felt that this material would not be interesting to most readers, so we put it in the appendix. In retrospect, perhaps that was a bad decision. At this stage in the paper's review cycle, we hesitate to move material around. We can, however, if the reviewer or editor insists.

---

## Author Comment (AC5) · 18 Mar 2018

We thank the reviewer for their comments. In this document, we detail our responses.

*1) It would be helpful to provide a little more physical motivation for the choice of tropical 500 hPa temperature. I see some good reasons why mid-tropospheric temperature should work better (e.g., it should scale better with LR, WV and LW cloud feedbacks), but I don't think this was discussed anywhere. Why use tropical temperature rather than global-mean? Is there a physical rationale, or did this simply work better in MPI-ESM?*

*Also, although mid-tropospheric temperature clearly works better for the overall feedback, I expect the scaling with Ta might actually be a worse choice for some individual feedback processes (e.g. surface albedo, marine low cloud). This might be worth discussing briefly.*

A: To address this, we have added a paragraph to the paper: "There are several plausible reasons why $T_A$ may control R better than $T_S$. It seems likely that several of the feedbacks — e.g., lapse rate, water vapor, longwave cloud — should be strongly influenced by atmospheric temperatures rather than $\Delta T_S$. More recently, it has been shown that atmospheric temperatures play a key role in regulating low clouds [Zhou et al., 2016, 2017], thereby influencing the shortwave cloud feedback. The net result is a clear dependence of ECS on atmospheric stability [Ceppi and Gregory, 2017]. We have not further investigated this — ultimately, our use of $\Delta T_A$ in Eq. 4 is based on empirical observations [Murphy, 2010; Spencer and Braswell, 2010; Trenberth et al., 2015] that it correlates well with $\Delta R$. Other metrics, such as global average atmospheric temperature work almost as well. Clearly, further investigations on how to best describe the Earth's energy balance are warranted."

*2) A key result is that the revised feedback parameter theta more accurately estimates the "true" feedback strength under CO2 forcing. This is shown to be the case in MPI-ESM (L172-176). However, does this hold for CMIP5 models in general? I.e., do the values of theta estimated in control runs correlate well with those in 4xCO2?*

A: This is not a claim we make in the paper, although one might infer it from the MPI model. Indeed, there is *some* correlation between short-term and long-term theta in the CMIP5 ensemble, as seen here:

[Figure]

Caption: Scatter plot of $\Theta_{4xCO2}$ vs. $\Theta_{control}$ from the CMIP5 ensemble. Each point represents values from model.

However, because of the outlier models, the relation is hard to interpret and we have not pursued this "emergent constraint" approach in our estimate of ECS using our revised framework [Dessler and Forster (2018, February 6). An estimate of equilibrium climate sensitivity from interannual variability. Retrieved from eartharxiv.org/4et67].

We have added a short statement to the paper to reflect this: "It may also be possible to use the relation between short-term and long-term Θ as an emergent constraint to convert short-term observations to the long-term response. There is some scatter in the relation in the CMIP5 ensemble, however, so more analysis of how these relate is likely required before ECS can be constrained in this way."

*Relatedly, I would also suggest adding the correlation between R and Ta in CMIP5 piControl to Fig. 4, as additional bars in a different color.*

A: We have done that.

*3) One important issue that isn't discussed in the paper is that the "pattern effect" doesn't simply go away with the improved relationship; rather, it shifts from the feedback parameter to the Ts/Ta term. This isn't a problem, but the way the paper is currently written, some readers might get that impression.*

A: We have added a sentence discussing this: "This means that the pattern effect's impact on ECS calculations shifts from $\lambda$ in Eq. 2 to the $\Delta T_S/\Delta T_A$ term in Eq. 4."

*So if most of the curvature in the relationship between radiative response and temperature goes away with the revised framework (Fig. 6), I expect there must be some curvature in the Ta versus Ts relationship in 4xCO2 runs. Can the authors confirm this?*

Confirmed.

[Figure]

Caption. Scatterplot of slope of $\Delta T_S$ vs. $\Delta T_A$ in CMIP5 abrupt4xCO2 runs. Each point represents one model. The dotted line is the 1:1 line. The subscripts (10-30, 30-150) indicate the years of the run from which the slopes are calculated.

We've added a sentence to the paper mentioning that there is curvature in $T_A$ vs $T_S$ relation: "One can conclude from this that there is curvature in the relation between $T_A$ and $T_S$ in the

models, emphasizing the need to improve our understanding of the factors that control $\Delta T_S/\Delta T_A$, including how future patterns of surface warming will evolve."

*4) I expect the Ts/Ta ratio cannot be reliably estimated from historical runs in the presence of large variability (for the same reason that lambda cannot be reliably estimated - because of the pattern effect). So we must rely on models to estimate this ratio under future global warming, meaning that it will be important to understand how future patterns of surface warming will develop. I suggest the authors discuss this briefly, for example in the conclusions.*

We have added a sentence to the paper mentioning this point: "One can conclude from this that there is curvature in the relation between $T_A$ and $T_S$ in the models, emphasizing the need to improve our understanding of the factors that control $\Delta T_S/\Delta T_A$, including how future patterns of surface warming will evolve."

*Other minor comments:*

*I suggest using colors in Fig. 6, rather than dark grey and black.*

Done

*L223: Cite Andrews and Webb 2018 - For future reference, it would be useful to mention the value of theta estimated from observations (horizontal dashed bar in Fig. 7a).*

Done.

---

## Author Response (AR1)

*Note that all line numbers in our responses refer to the version with tracked comments included in this document. Line numbers from the reviewers refer to the original manuscript.*

Reviewer #1

We thank the reviewer for their comments. Below, we detail our responses.

*64-65. As a single number to quantify the spread, the standard deviation would also be helpful.*

We have added 5-95% confidence intervals throughout the paper.

*66. Why do you use only a single decade, rather than all the data, for instance by dividing the dataset into two or using regression (cf Barnes and Barnes, 2015, 10.1175/JCLI-D-15-0032.1)? A single decade would be less precise. You could estimate the statistical uncertainty incurred from the control run.*

We calculate ECS using this approach because this is the way most ECS calculations based on the 20[th]-century observational record are done. Thus, our results can therefore directly provide insight into the impact of variability in the observational estimates of ECS.

The reviewer is correct that using more than a decade might affect the results. If one used the difference between the averages of the first and last 20 years, the range in $\lambda$ declines from 0.87 W/m$^2$/K to 0.48 W/m$^2$/K. Using longer averaging periods does not further decrease the range. We now mention this in the paper (line 67).

*118. It would be useful to remark here that 16 years is chosen to match the CERES dataset, because that was mentioned some lines above (103-104), where it appears actually to be 17 years and 5 months long.*

We have added a statement that the segmentation of the data is done to match the CERES record (line 134). We have also updated the paper to segment the data into 17-year segments to more closely match CERES.

*119, 196. Why are monthly anomalies used here, rather than annual? Does it make a difference?*

We do this to facilitate the comparison with the CERES regressions, which also uses monthly data. The reason most analyses with CERES data are done with monthly data is because using annual data means there's only 17 data points, and the uncertainties end up being very large. Issues involved in annual vs. monthly regressions are discussed in some detail in Forster (2016, 10.1146/annurev-earth-060614-105156).

*167. Again, the standard deviation would be helpful, and could be compared with lines 64-65.*

Added.

*173, 175. You could give standard errors of the mean for each of these two numbers, and judge the significance of their difference.*

We have added the 5-95% confidence intervals to all of these numbers.

*174, 175. "analysis" and "calculated" - by what method? From the slope of R against Delta T?*

We have clarified the text that we use the method of Gregory et al. (2004), where annual average R is regressed against T, and the slope of the curve is an estimate of λ or Θ (line 194)

*204. "agrees" in what sense?*

We have changed the sentence to read: "We find that the 15 models whose average short-term Θ falls within the uncertainty of Θ estimated from CERES observations have ECS values ranging from 2.0-3.9 K, with an average of 2.9 K." (line 247)

*218. I would say that this is "one source" of the spread, which is not eliminated, but only reduced, by using Theta instead.*

We believe that this sentence is phrased correctly. The spread in our estimate from the ensemble is due to the construction of the energy-balance equation. Unlike observational analyses, we know everything else perfectly. Using our revised energy balance equation does not completely solve the problem, but it is an improvement.

*233. Why is this material an appendix, rather than being incorporated in the main text?*

We felt that this material would not be interesting to most readers, so we put it in the appendix. In retrospect, perhaps that was a bad decision. At this stage in the paper's review cycle, we hesitate to move material around. We can, however, if the reviewer or editor insists.

Reviewer #2

We thank the reviewer for their comments. In this document, we detail our responses.

*1) It would be helpful to provide a little more physical motivation for the choice of tropical 500 hPa temperature. I see some good reasons why mid-tropospheric temperature should work better (e.g., it should scale better with LR, WV and LW cloud feedbacks), but I don't think this was discussed anywhere. Why use tropical temperature rather than global-mean? Is there a physical rationale, or did this simply work better in MPI-ESM?*

*Also, although mid-tropospheric temperature clearly works better for the overall feedback, I expect the scaling with Ta might actually be a worse choice for some individual feedback processes (e.g. surface albedo, marine low cloud). This might be worth discussing briefly.*

A: To address this, we have added a paragraph to the paper beginning on line 221.

*2) A key result is that the revised feedback parameter theta more accurately estimates the "true" feedback strength under CO2 forcing. This is shown to be the case in MPI-ESM (L172-176). However, does this hold for CMIP5 models in general? I.e., do the values of theta estimated in control runs correlate well with those in 4xCO2?*

A: This is not a claim we make in the paper, although one might infer it from the MPI model. Indeed, there is *some* correlation between short-term and long-term theta in the CMIP5 ensemble, as seen here:

[Figure]

Caption: Scatter plot of $\Theta_{4xCO2}$ vs. $\Theta_{control}$ from the CMIP5 ensemble. Each point represents values from model.

However, because of the outlier models, the relation is hard to interpret and we have not pursued this "emergent constraint" approach in our estimate of ECS using our revised framework [Dessler and Forster (2018, February 6). An estimate of equilibrium climate sensitivity from interannual variability. Retrieved from eartharxiv.org/4et67].

We have added a short statement to the paper to reflect this on line 260: "It may also be possible to use the relation between short-term and long-term $\Theta$ as an emergent constraint to convert short-term observations to the long-term response. There is some scatter in the relation in the CMIP5 ensemble, however, so more analysis of how these relate is likely required before ECS can be constrained in this way."

*Relatedly, I would also suggest adding the correlation between R and Ta in CMIP5 piControl to Fig. 4, as additional bars in a different color.*

A: We have done that.

*3) One important issue that isn't discussed in the paper is that the "pattern effect" doesn't simply go away with the improved relationship; rather, it shifts from the feedback parameter to the Ts/Ta term. This isn't a problem, but the way the paper is currently written, some readers might get that impression.*

A: We have added a sentence discussing this: "Thus, the pattern effect's impact on ECS calculations shifts from $\lambda$ in the traditional framework to the $\Delta T_S/\Delta T_A$ term in Eq. 4." (line 217)

*So if most of the curvature in the relationship between radiative response and temperature goes away with the revised framework (Fig. 6), I expect there must be some curvature in the Ta versus Ts relationship in 4xCO2 runs. Can the authors confirm this?*

Confirmed.

[Figure]

Caption. Scatterplot of slope of $\Delta T_S$ vs. $\Delta T_A$ in CMIP5 abrupt4xCO2 runs. Each point represents one model. The dotted line is the 1:1 line. The subscripts (10-30, 30-150) indicate the years of the run from which the slopes are calculated.

We've added a sentence to the paper mentioning that there is curvature in $T_A$ vs $T_S$ relation: "The lack of curvature in the $\Theta$ calculations means there is curvature in the relation between $T_A$ and $T_S$ in the models." (line 216)

*4) I expect the Ts/Ta ratio cannot be reliably estimated from historical runs in the presence of large variability (for the same reason that lambda cannot be reliably estimated - because of the pattern effect). So we must rely on models to estimate this ratio under future global warming, meaning that it will be important to understand how future patterns of surface warming will develop. I suggest the authors discuss this briefly, for example in the conclusions.*

We have added a sentence to the paper mentioning this point: "This also emphasizes the need to improve our understanding of the factors that control $\Delta T_S/\Delta T_A$, as well as how future patterns of surface warming will evolve." (line 218)

*Other minor comments:*

*I suggest using colors in Fig. 6, rather than dark grey and black.*

Done

*L223: Cite Andrews and Webb 2018 - For future reference, it would be useful to mention the value of theta estimated from observations (horizontal dashed bar in Fig. 7a).*

Done.

[revised manuscript text omitted]